# The Relationship Between the Burpee Movement Program and Strength and Endurance Performance Measures in Active Young Adults: A Cross-Sectional Analysis

**DOI:** 10.3390/jfmk9040197

**Published:** 2024-10-18

**Authors:** Ľuboslav Šiska, Gheorghe Balint, Daniel Židek, Jaromir Sedlacek, Štefan Tkacik, Nela Tatiana Balint

**Affiliations:** 1Department of Physical Education and Sports, Catholic University in Ružomberok, 03401 Ruzomberok, Slovakia; daniel.zidek@ku.sk (D.Ž.); jaromir.sedlacek@ku.sk (J.S.); 2Faculty of Movement, Sports and Health Sciences, University Vasile Alecsandri of Bacau, 600115 Bacau, Romania; tbalint@ub.ro; 3Department of Informatics, Catholic University in Ružomberok, 03401 Ruzomberok, Slovakia; stefan.tkacik@ku.sk

**Keywords:** sport science, motor performance, mobile app, burpee, intensity, index of fatigue

## Abstract

Objectives: This paper aimed to assess the motor performance in the Burpee Movement Program through the acceleration recorded by the Phyphox mobile app and define its relationship to strength and endurance parameters. Methods: Altogether, 15 students in physical education teaching completed the 3 × 3 min Burpee Movement Program, consisting of the repeated execution of a single burpee with maximum effort at regular intervals triggered by a sound signal. During the load phase, the intensity of the burpee and the fatigue index expressed in percentages were evaluated by means of the acceleration recorded through a mobile phone. In the second part of testing, we evaluated the performance parameters during a bench press and squat where the intensity was measured using a linear displacement transducer (Tendo Power Analyzer) and aerobic endurance was assessed with a 20 m shuttle run test (20 mSR). Results: The average intensity of the burpee ranged from 3.12 to 11.12 ms^−2^. The fatigue index ranged from −21.95% (which represented an increase in performance) to 33.63% (which represented a decrease in performance). The performances in the bench presses ranged from 58 to 480 W and from 175 to 696 W during the squats. The distance in the 20 m shuttle run test (20 mSR) ranged from 540 to 2000 m. The intensity of the burpee showed a significant correlation to the performances achieved in the bench presses and squats r = 0.82 and 0.79. The fatigue index showed a significant correlation to the 20 m shuttle run test (20 mSR) r = −0.67. Conclusions: These findings indicate that in, our case, the results from the Burpee Movement Program are significantly associated with the participants’ strength and endurance abilities. We recommend using BMP for the development of strength–endurance abilities, but further exploration is needed regarding the potential use of BMP as a diagnostic test.

## 1. Introduction

Sports have been evolving and innovating since time immemorial, whether in the form of improved sports equipment, various training methods or the use of modern technologies. In terms of training methods, we can observe the increasing popularity of high intensity interval training (HIIT) programs, which combine load intervals performed at a high intensity with intervals of rest [1,2,3,4,5]. They can also be performed using many types of equipment and exercises, such as bicycles, treadmills, running, free weights and sailor’s ropes, as well as exercises that use one’s own body weight as a means of resistance [6,7,8]. When discussing the integration of modern technology into sports, mobile phones are now utilized in various ways during physical activities. One of the simplest forms is through apps that offer different timers, whether for Tabata, HIIT workouts, or more advanced tools simulating standardized tests like the Leger beep test or the yo-yo intermittent test [9]. With the global positioning system (GPS) included in mobile phones, it is also possible to track running speed, distance and some other parameters [10]. Recently, the accelerometer feature in smartphones has become increasingly valuable, whether for tracking daily activities [11], counting exercise repetitions like squats, push-ups, or burpees and recording them [12,13], or even estimating the intensity of physical activity [14].

Our previous research is a combination of these areas. We designed several short movement programs, and we were inspired by the works of Oeurgui [15] and Hatfield [16], who created a program that had the same duration as a boxing and kickboxing match, which we also used in our case. However, the content of our program was more oriented towards the HIIT principles. We used different types of exercises in them, either with the participants’ own weight, as with the modified burpee [17,18], the barbell or short sprints [4,19], but their content always revolved around the repeated performance of given exercises that could be quantified numerically at certain time intervals. To measure these exercises, we used various devices including Fitro Agility (FITRONIC, Bratislava, Slovakia), a photoelectric cell system (Microgate, Bolzano, Italy), and a linear displacement transducer (Tendo Sports Machines, London, UK). This allowed us to collect a set of data throughout the program. The number of values obtained during the programs ranged from 30 for short sprints to 78 for the modified burpee. Through a linear trend, we were able to identify whether the performances during the program increased or decreased. This provided us with two basic parameters: an immediate performance index and a fatigue index expressed as the percentage difference between the maximum and minimum values on the linear trend line regarding the minimum value. However, the above programs suffered from certain limitations, either in terms of their necessary measuring devices or their material aids, which forced us to simplify them to the current extent. We presented the Burpee Movement Program (BMP) [20], which consisted of repeating burpees in accordance to a sound signal at regular intervals. Burpees have been used in many studies, either as part of the development of a movement performance, or as part of a diagnostic test, but only in terms of the number of repetitions [21,22,23,24,25,26]. The methodology of performing this exercise varied: in some cases, a jump was not performed, or the chest had to touch the ground, etc. In our research, we decided that the participants would perform the burpee with a jump, but the subjects did not have to perform push-ups during the exercise. This option was chosen because of its speed, which was our focus. When designing the BMP, we tried to make it modifiable both in terms of volume and intensity. Using a mobile phone and the PHYPHOX app, we managed to record the course of acceleration during a burpee [27], and based on the works of Lee [28], Quiroz [29] and Straczkiewicz [30] where different ways of capturing the physical activity using a mobile phone were described, we developed an algorithm for detecting the values of acceleration that make up one repetition of the exercise, and by averaging them, we expressed the intensity of the burpee in meters per second squared, which is the base unit of acceleration. This allowed us to monitor the performances during the program using a simple and accessible device, and evaluate the intensity of a single burpee repetition, which, considering the nature of the exercise, can represent the participant’s strength potential and the fatigue index in percentages, which, depending on the length of the program, can represent the participant’s endurance potential [18].

Understanding the relationships between different fitness parameters is essential for designing effective training programs adapted to individual needs and goals. If we assert that the Burpee Movement Program can represent both strength and endurance potential, it is crucial to understand the relationship between these abilities. This assessment is crucial before using this instrument in research contexts. From the perspective of strength abilities, Soriano [31,32] asserts that the fundamental exercises for strength development are squats and bench presses which, with the performance expressed as the product of barbell weight and execution speed, can serve as an indicator of strength abilities. This approach was used in research studies by Zemková [33,34]. On the other hand, Ramsbottom [35] and Magee [36] suggest that maximum oxygen uptake values can be predicted from the level achieved in a 20 m progressive shuttle run test, which can also provide insight into the endurance running ability of active individuals. These parameters appear suitable for comparison, which leads to the objective of our study. The aim of our research was to assess the motor performances in the Burpee Movement Program through acceleration recorded by the Phyphox mobile app 1.1.16 and to define the relationships with strength and endurance parameters. We hypothesize that the intensity of a single burpee is dependent on strength parameters, while the fatigue index correlates with the level of endurance.

## 2. Materials and Methods

### 2.1. Experimental Approach to the Problem

A cross-sectional correlation analysis study was used to investigate the association between the performances in the BMP and the performances in strength and endurance tests in the students of physical education. The participants were tested in December (2023) over three days with 48 h of rest between (Figure 1). All the tests were performed indoors and at the same time of the day. A 15 min warm-up consisting of low-intensity jogging, dynamic stretching, running drills and task-specific high-intensity activities was completed before the tests. The participants were allowed to perform low-intensity physical activity between the testing days.

### 2.2. Participants

In our research we used a non-probability convenience sampling method based on non-random criteria. The participants were selected from bachelor’s degree students of physical education teaching, and we collected data from 15 students (consisting of 11 men and 4 women), who completed all three testing sessions. Of the total number of 23 students who participated in testing session 1, one was excluded due to problems with their mobile device. Subsequently, three students did not pass testing session 2 and four others did not pass testing session 3. As part of the educational process to obtain a degree in physical education, the burpee exercise was regularly included, so the participants had no issues performing it. All the participating students had a sufficient level of physical fitness to complete the BMP. The participants did not suffer from orthopedic and neurological injuries. After being informed about the experimental procedures, their risks and benefits, all the subjects agreed to participate voluntarily and signed a written informed consent form before data collection. This study was approved by the Ethics Committee of the Vasile Alecsandri University of Bacău (Nr. 31/1/02.09.2024) with respect to the ethical standards of the Declarations of Helsinki.

Anthropometry measurements were taken at the beginning of testing session 1, using conventional criteria and measuring procedures. The body weights (BWs) were assessed to the nearest 0.1 kg using a certified electronic scale (Tanita electronic scale BWB-800 MA (Wunder SA.BI. Srl, Trezzo do Ada, Italy)). The body heights (BHs) were measured to the nearest 0.1 m using a Harpenden portable stadiometer (Holtain Ltd., Crymych, Pembs. UK). The body mass indices (BMIs) were calculated as kg/m^2^ (Table 1).

### 2.3. Procedures

#### The Burpee Movement Program (BMP)

The test was identical to the timing of a boxing match, i.e., 3 **×** 3 min, with a one-minute break between the rounds. The participant performed 26 single burpees at regular intervals to a sound signal during a 3 min round. The course of a single burpee consisted of the basic upright standing position (1), from which the participant moved after the sound signal into squatting (2) and then the lying position (3) with outstretched or slightly bent arms. In the shortest possible time, followed by the transition from the lying position to squatting (4) into a jump (5) and back to the basic position (6) (Figure 2).

The principle of the test is as follows: the participant is in the basic position and waits for a sound signal (beep), immediately after which they perform a single burpee, remain in that basic position and wait for the next beep, which repeats every 7 s. In this way, the participant performs 26 single burpee repetitions, while the last signal in the round is at 2:55 min (175 s) followed by a 1 min break. In total, in three rounds, 78 single burpee repetitions are performed in this way. The participants were instructed to perform the repetitions as quickly as possible to maintain the resting phase without significant activity between the repetitions and to not put their hands above their head when jumping [20].

We used a mobile phone, which can capture the measurement data using its sensors, including the built-in accelerometer, to measure the activity during the short-interval exercise. The acceleration values were collected using the PHYPHOX mobile app (RWTH Aachen University, Aachen, Germany) with the mobile phone in a case attached to the test subject’s left arm just below the deltoid muscle. The PHYPHOX app recorded the accelerations along the x, y and z axis and the total magnitude of acceleration √ (x^2^ + y^2^ + z^2^) in ms^−2^ with the relevant time stamps. The values were recorded without the gravity G component and when the mobile device was motionless, the acceleration values had a zero value in all the positions. In our research, an iPhone SE with a recording frequency of 100 Hz was used. The measurement data were exported in the csv format to Microsoft Excel Microsoft 365 subscription (Microsoft Corporation, Redmond, WA, USA).

The heart rate was recorded using a SUUNTO device (Suunto, Vanta, Finland), which receives wireless signals from the Suunto heart rate monitor. It displays the heart rate of athletes on the monitor in real time, captures the changes every second and provides information about each athlete’s performance. The SUUNTO team pack device consists of a PC, the Suunto receiver and the chest belt. The maximum heart rate values in the individual rounds (Hrmax1, 2, 3) were used in the results section.

### 2.4. Power Performance

The assessment of the power parameters was carried out through a series of diagnostic bench presses and squats. The diagnostic series started with a barbell weight of 20 kg, which gradually increased by 5 kg, until the participant was unable to perform the attempt or did not perform the attempt at the next weight by his/her own decision. The barbell was secured at every moment by a pair of assistants on the sides. At each weight, the participant performed one repetition with a rest interval of at least 3 min between the individual repetitions. The intensity of each repetition was measured and recorded using a linear displacement transducer Tendo Power Analyzer (Tendo sports machines UK Ltd.), which has been demonstrated to be a dependable and accurate tool for power measurement [37,38]. The device was attached to the barbell perpendicular to the floor using a nylon cord, allowing real-time measurement. The Tendo Power Analyzer was connected to the computer with compatible software (Tendo sports machines UK Ltd.), providing immediate feedback on barbell velocity (m/s) and power (W). The average performance in the concentric phase of each subject’s exercise was used in the results part (average performance bench press—avBP, average performance squat—avSQ).

The exercises were conducted following the standards defined by the International Powerlifting Federation (International Powerlifting Federation, 2019) [39]. All the participants completed a familiarization session. They were introduced to the proper technique for performing the bench press and squat both with and without added weight. Each test subject had the opportunity to practice the exercises. Moreover, all the test subjects were already acquainted with the devices used in the study and the measurement methodology.

During the bench press, the participants were laying on a weight-bench with five points of contact—the head, buttocks and both shoulders were on the bench, and both feet were flat on the ground. The participants took the barbell from the rack and held it with their arms extended. When the investigator issued the “start” command, the participants lowered the barbell until it contacted their chest and then pressed it upwards until their arms were fully extended. A “rack” command was then issued to return the barbell to the rack. In the squat exercise, the participants started in an upright position with their hips and knees fully extended, and the barbell was positioned across their upper back or shoulders. Upon the investigator’s “squat” command, the participants squatted until their hip joint moved below the knee line. Afterwards, the participants returned to the starting position. The “rack” command was then issued to return the barbell to the rack. In accordance with Macarilla [40], to maintain ecological validity, no other commands were provided.

### 2.5. Endurance Performance

Aerobic endurance was assessed with a 20 m shuttle run test (20 mSR). According to Ramsbottom [35], the test consisted of repeated 20 m shuttles performed at increasing speeds until exhaustion. Audio cues were recorded on a CD. The test subjects were required to complete as many shuttles as possible. The test was considered completed when the test subject was no longer able to follow the specific pace for 2 successive shuttles or stopped because of exhaustion. The total distance covered was recorded (beep) [41]. The heart rate during the test was recorded in the same way as was done for BMP. The maximum value achieved during the test (Hrmax beep) was used in the results section.

### 2.6. Data Processing

In our research, we worked with the values of total acceleration magnitude (mag). The Phyphox mobile app recorded around 66,000 readings during the load phase. One burpee was represented by approximately 200 to 400 acceleration values depending on the execution speed. When processing the data from the accelerometer, we extracted the values for each burpee separately in MS Excel and determined their average value. When filtering the values representing the burpee, we used the moving average smoothing method. To eliminate the unwanted external factors, an average of 11 values (5 before and 5 after the given value) was used instead of the given value, and the data were smoothed 9 times with this method. The beginning of the exercise was identified by the smoothed acceleration value exceeding the limit of 0.5 and not falling below this limit for the next 200 values. The end of the exercise was identified in a similar way: the acceleration value had to exceed the threshold of 3 and not fall below this threshold for the next 200 values [27]. We calculated the average from the selected single repetition data, which indicated the intensity of the burpee. This way, we obtained 26 values for one round. The results included the average values of the burpees in individual rounds (AIB1, 2, 3), where 1, 2 and 3 meant the first, second and third round, respectively, as well as in the entire exercise program (AIB). The performance drop is expressed as the percentage difference between the maximum and minimum values on the linear trend line regarding the minimum value. When the performance decreased, the linear trend line was dropping with a positive value in the fatigue index, and when the performance increased during the round, the linear trend line was rising with a negative value in the fatigue index. The results included the fatigue index in the individual rounds (IF1, 2, 3) as well as in the entire exercise program (IF).

### 2.7. Data Analysis

The data are reported as mean ± standard deviation (SD). The Shapiro–Wilk test for normality was carried out on all the variables. The Student paired samples *t*-test was used to compare the values between the three rounds in the burpee movement program and to compare the differences between bench press and squat performances, and the maximum heart rate in the burpee movement program and the 20 m shuttle run test. Pearson’s correlation (r) was used to determine the relationships between the variables of individual tests. The magnitude of the correlation coefficients was analyzed according to small (0.1 to 0.29), moderate (0.3 to 0.49), large (0.5 to 0.69), very large (0.7 to 0.89), and extremely large (0.9 to 1). The significance level was at *p* < 0.05. Statistical analyses were performed in MS Excel 2016 (Microsoft corporation, Redmond, WA, USA) and JASP 0.16.4.0 (Department of Psychological Methods University of Amsterdam, The Netherlands).

## 3. Results

When evaluating this specific program, we can state that the intensity of the burpee ranged in the first round from 9.55 ms^−2^ to 12.53 ms^−2^ (represented by bars, with one bar representing the intensity of one burpee repetition) with an average intensity of 11.04 ms^−2^. The linear performance (represented by a dotted line) increased from 10.82 ms^−2^ to 11.25 ms^−2^, which represented a negative fatigue index and performance growth at a level of 4.00%. In the second round, the intensity of the burpee ranged from 10.06 ms^−2^ to 13.00 ms^−2^ with an average intensity of 11.65 ms^−2^. The linear performance dropped from 11.95 ms^−2^ to 11.34 ms^−2^, which represented a fatigue index of 5.09%. In the third round, the intensity of the burpee ranged from 6.09 ms^−2^ to 12.43 ms^−2^ with an average intensity of 10.66 ms^−2^. The linear performance dropped from 12.20 ms^−2^ to 9.13 ms^−2^, which represented a fatigue index at a level of 25.17%. When evaluating the entire course of the test, the average intensity of the burpee in all three rounds (average of all 78 repetitions) was at a level of 11.12 ± 1.25 ms^−2^. The linear performance (represented by a solid line) dropped from 11.55 ms^−2^ to 10.69 ms^−2^, which represented a fatigue index of 7.45% (Figure 3).

When monitoring the execution of the Burpee Movement Program within the entire set, a decrease in the average intensity of the burpee between the rounds was observed; the decrease between the first and second round was not statistically significant (t(14) = 1.3, *p* = 0.20), the decrease between the first and third round was significant (t(14) = 2.78, *p* = 0.01) and the decrease between the second and third round was also significant (t(14) = 2.23, *p* = 0.04). The fatigue index dropped from the first round to the second but not significantly (t(14) = 1.33, *p* = 0.21) and it increased from the first to the third round, but not significantly (t(14) = −1.69, *p* = 0.11); only the increase from the second to the third round was significant (t(14) = −3.31, *p* = 0.00). The heart rate increased significantly between the first and second round (t(14) = −10.49, *p* = 0.00), the first and third round (t(14) = −9.72, *p* = 0.00), and second and third round (t(14) = −4.52, *p* = 0.00).

The values of the average performance during the bench press ranged from 60 to 480 watts, which ranged from 0.92 to 6 w/kg^−1^ when converted to kilograms of weight. During the squat, the average power was 175 to 696 watts, which ranged from 2.62 to 8 w/kg^−1^ when converted to kilograms of weight. The performance in the squat exercise was significantly higher than in the bench press exercise (t(14) = −9.11, *p* = 0.00). The performance in the shuttle run varied from 540 to 2000 m at a maximum heart rate from 173 to 203 bpm. The maximum heart rate during the beep test was significantly higher than during the BMP (t(14) = 6.06, *p* = 0.00). (Table 2).

When monitoring the relationships between the individual parameters, we can see a high consistency between the average burpee intensity in the round and the overall intensity for the entire BMP at a level from 0.96 to 0.99. The fatigue index did not show a similar consistency and one of the relationships between IF1–IF3 was not significant, even though the relationships were mostly significant (Table 3).

The average performance in the concentric phase during the bench presses and squats significantly determined the intensity of the burpee, but so did the performance in the beep test, which, as a manifestation of endurance, showed a significant relationship with the BMP fatigue index. The strongest relationship was revealed between the average power in the bench press and squat, and we also noted significant relationships between these parameters and the average performance in the beep test (Figure 4).

## 4. Discussion

The primary aim of this study was to evaluate specific motor performances during the Burpee Movement Program (BMP), focusing on both the intensity of a single burpee and the fatigue index over the course of the test. Additionally, we aimed to explore the relationships between BMP performance and the key indicators of strength and endurance abilities. The findings revealed a very large correlation between single burpee intensity and strength parameters. In examining the interrelationships between fitness components, the connection between explosive strength and speed abilities becomes evident, consistent with previous studies [31,32], which identified significant relationships between these variables. Similarly, it was shown that the maximal dynamic back squat performance significantly influenced agility [42]. This suggests that single burpee intensity, which is predominantly influenced by strength, could enhance speed performance under specific conditions. A noteworthy observation regarding burpee intensity is the critical role of upper limb strength. We infer that the plank phase of the burpee, where the subject supports their body weight, is a key determinant of performance due to the high demand on the upper limbs. Considering that upper limb strength significantly impacts punching power [43], this aspect can be highly beneficial in combat sports. But, this also underscores the importance of upper body strength training, particularly in sports like basketball and volleyball, where upper limb power and stability are essential. In our previous research [20], we also established a very large correlation between burpee intensity and vertical jump height, further indicating that burpees are a full-body exercise that engages both upper and lower body muscle groups. This engagement enhances overall power output and neuromuscular coordination. Moreover, we found a significant relationship between the fatigue index and the participant’s endurance level. The fatigue index, often employed as a diagnostic measure of anaerobic capacity in Wingate tests (defined as a drop in the performance on a cycling ergometer over 30 s in watts) [44], showed no significant association with aerobic endurance, as confirmed by other recent studies [45,46], such as those concerning repeated sprint ability. Of course, the length and course of the test differ significantly from BMP. The fatigue index calculation in this study was based on the percentage difference between the best and worst performances across multiple test trials. Rather than using absolute values for each trial, we adopted a linear trend approach, treating the best and worst performances as endpoints within the overall series. This methodology provided a more comprehensive assessment of fatigue accumulation throughout the exercise. Focusing on the fatigue index from the graph presented in the results section, it might seem that the noticeable fatigue occurred only during the last four repetitions of the program. However, this was an isolated case as, for other participants, the decline in performance was evident throughout the entire program. In some cases, particularly among physically fitter participants, we even observed an increase in performance. The analysis of fatigue indices revealed significant differences across individual trials, which prompted a final evaluation using an overall fatigue index that includes all three test rounds. It is important to consider that motivation plays a crucial role in influencing the participant’s maximum performance. The lower correlation between the fatigue index and endurance capacity in some cases may be attributed to an insufficient test load for certain individuals, leading to slightly skewed results. However, the BMP is modifiable in terms of the number of single burpee repetitions, which could eliminate the aforementioned issue. In the variant with a 5 s interval, the fatigue index was noticeable throughout the entire BMP process. A surprising finding was the significant relationship between strength parameters and endurance levels, which we attribute to the size and composition of the research sample.

Focusing on the physiological response during BMP, in a study by Oeurgui [15], which featured a similar exercise duration, maximum heart rates of up to 190 bpm were recorded and similar values were observed by Moura [21] during the implementation of an adapted burpee test. This is comparable to our findings, where participants also approached near-maximal heart rates during the BMP test. According to the research by Weltman [47], heart rates at 90–95% of the maximal level correspond to the lactate threshold, suggesting that our BMP protocol operates within this physiological interval. The data collected in our current study are in line with previous findings. The burpee intensity ranged from 3.12 ms^−2^ in the worst case to 11.65 ms^−2^ in the best case. In earlier studies, the intensity during slow burpee execution was around 3.50 ms^−2^, while the maximal effort yielded 12.92 ms^−2^. The slightly higher values in earlier research could be attributed to the fact that participants regularly performed BMP exercises, leading to greater adaptation. Similarly, the fatigue index in our study ranged from negative values, indicating performance improvement, to positive values, which reflected performance decline. In conclusion, the BMP can serve as a highly effective training tool. Without the necessity for specialized equipment, it offers a simple yet versatile exercise regime. BMPs can be effectively incorporated as supplementary elements within a broader training framework and, in some cases, may even serve as a central component. Given the total time required for a BMP session, including warm-up and cool-down stretching, it can comfortably fit within a 45 min workout. This makes the BMP particularly suitable for use as a fitness module in physical education and sport training programs. On the other hand, the ability to assess intensity during the BMP opens up possibilities for its use as a test of strength–endurance abilities, although this needs to be thoroughly verified.

## 5. Conclusions

We can conclude that we managed to define the relationships between specific movement performance in the BMP and standardized strength and endurance tests. The intensity of burpees showed a significant relationship with the average performance in the concentric phase for the squats and bench presses, and the fatigue index showed a significant relationship with the endurance performance in the 20 m shuttle run test. The BMP results can be used to predict the level of strength and endurance. However, these findings cannot be generalized, mainly due to the size of the research sample. We recommend using BMP for the development of strength–endurance abilities, but further exploration is needed regarding the potential use of BMP as a diagnostic test.

## Figures and Tables

**Figure 1 jfmk-09-00197-f001:**
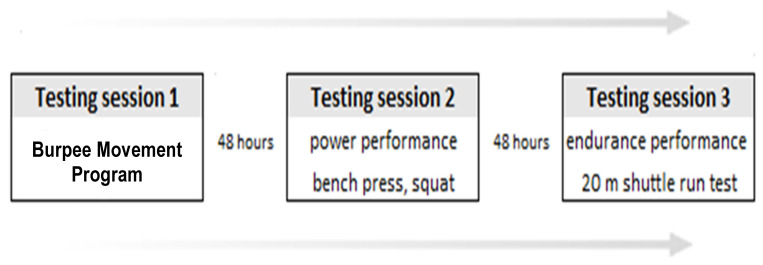
Testing protocol.

**Figure 2 jfmk-09-00197-f002:**
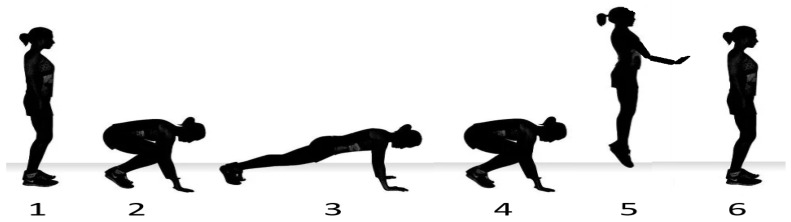
Phases of single burpee.

**Figure 3 jfmk-09-00197-f003:**
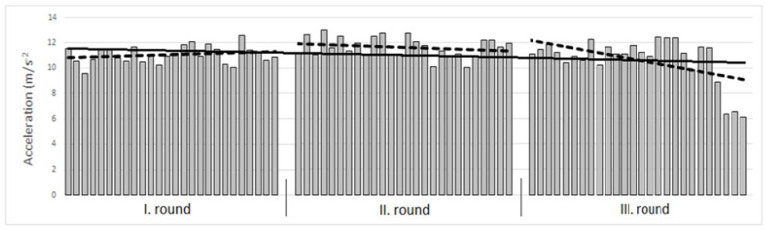
Execution of the Burpee Movement Program of an individual participant.

**Figure 4 jfmk-09-00197-f004:**
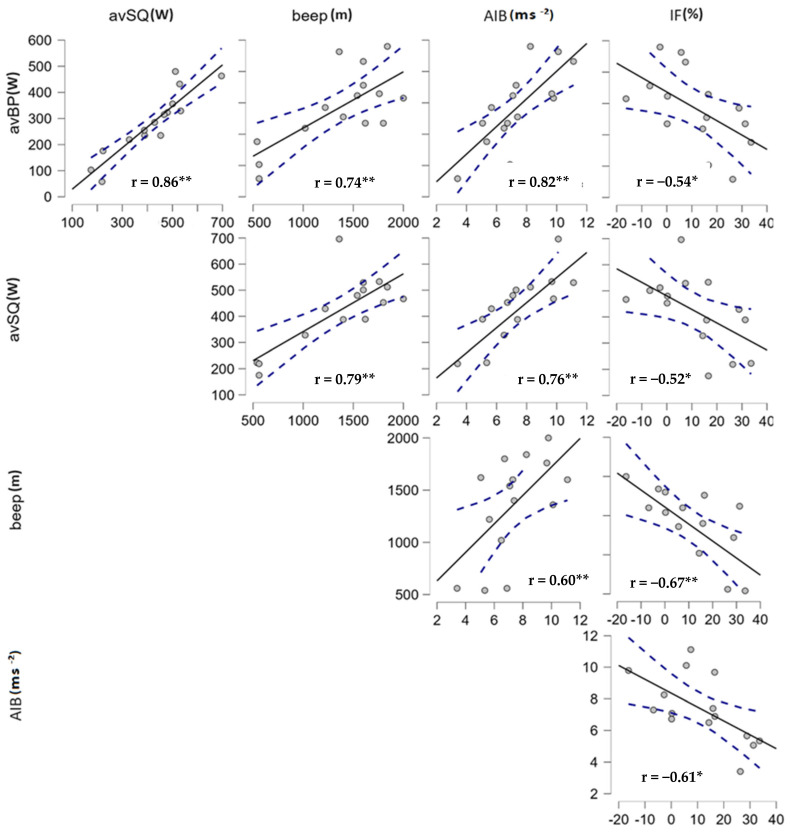
Course of correlation relations of the monitored variables, *—*p* < 0.05; **—*p* < 0.01.

**Table 1 jfmk-09-00197-t001:** Descriptive statistics of the participants.

	M	SD	Min	Max
Age	22.53	5.50	19	31
BH	179.93	10.29	162	202
BW	74.60	13.39	50	95
BMI	22.88	2.58	19.05	28.08

**Table 2 jfmk-09-00197-t002:** Values of the parameters in the Burpee Movement Program.

	P-Shapiro–Wilk	Mean	SD	Min	Max	*t* Test
AIB (ms^−2^)	0.85	7.35	2.12	3.40	11.12	
AIB1	0.66	7.56	1.94	3.66	11.04	1–2 N.S.
AIB2	0.78	7.38	2.27	3.43	11.65	1–3 *
AIB3	0.88	7.10	2.22	3.12	10.66	2–3 *
IF (%)	0.76	11.42	14.87	−16.28	33.63	
IF1	0.71	6.71	11.60	−9.33	30.02	1–2 N.S.
IF2	0.43	3.8	10.75	−21.46	21.84	1–3 N.S.
IF3	0.05	12.48	12.99	−21.95	28.96	2–3 **
avBP (W)	0.88	284.27	121.72	60.00	480.00	BP–SQ **
avSQ (W)	0.47	421.47	139.51	175.00	696.00	
Beep (m)	0.06	1361.33	485.35	540.00	2000.00	
Hrmax1 (bpm)	0.42	164.20	10.62	150.00	183.00	1–2 **
Hrmax2	0.22	170.67	9.69	156.00	184.00	1–3 **
Hrmax3	0.09	173.47	10.05	155.00	185.00	2–3 **
Hrmaxbeep	0.16	188.13	6.86	173.00	203.00	Hr bpm—beep **

Notes: AIB—average intensity burpee; IF—index of fatigue; Hrmax—maximum heart rate; 1, 2, 3—first, second, third round; avBP—average performance bench-press in watts; avSQ—average performance squat in watts; Beep—performance in 20 m shuttle run test; Hrmaxbeep—maximum heart rate during 20 m shuttle run test. N.S.—nonsignificant; *—*p* < 0.05; **—*p* < 0.01.

**Table 3 jfmk-09-00197-t003:** Consitency of BMP parameters in individual rounds.

	AIB1	AIB2	AIB3		IF1	IF2	IF3
AIB2	0.98 **			IF2	0.55 *		
AIB3	0.96 **	0.98 **		IF3	0.43	0.59 *	
AIB	0.99 **	0.99 **	0.99 **	IF	0.80 **	0.55 *	0.72 **

Notes: AIB—average intensity burpee; IF—index of fatigue; Hrmax—maximum heart rate; 1, 2, 3—first, second, third round, *—*p* < 0.05; **—*p* < 0.01.

## Data Availability

The data presented in this study are available upon reasonable request from the corresponding author.

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
