# Peer review of "The Relationship Between the Burpee Movement Program and Strength and Endurance Performance Measures in Active Young Adults: A Cross-Sectional Analysis"

_jfmk, 2024, doi:10.3390/jfmk9040197_

Round 1

Reviewer 1 Report

Comments and Suggestions for Authors

This is the reporting of a research studying the association between performance metrics of the Burpee Movement Program (BMP) and performance metrics of popular field strength/power and cardiorespiratory endurance measures. I find the article a little hard to read, either because authors are not clear when presenting their rationale, or because there are a lot of abbreviations that are not defined (e,g, GPS, HIIT), particularly in figures and tables in the Results section, hindering readability, or statistical analysis are not clear, e.g., results of ANOVAs are shown in Results but which models have been used, including which post hoc tests, have not been reported in Data Analysis section. Considering the study objectives, in addition, authors don’t clearly discuss the best correlations between BPM metrics and the other tests performance and why they best associate. They discuss a lot of correlations between other performance tests which is not their main study goal.

Title: Your title is a poorly informative. I suggest something like: “The Burpee Movement Program test and power and endurance performance measures: a cross-sectional correlation study in active young adults” or “The association between the Burpee Movement Program test and power and endurance performance measures in active young adults: a cross-sectional analysis”

Abstract: Clearly state your study objectives, i.e., refer to the burpee performance. Methods are incomplete. In results, if the Leger beep test is the 20m shuttle run test, just say is the 20m shuttle run test, otherwise you may confound several readers. Don’t use alternative names unless you clearly inform the reader on that (even so, avoid it for simplicity purposes).

Introduction

Line 46: Cite your studies accordingly in the following sentences. Otherwise, it appears that you are only self-citing your studies.

Line 59: When you refer “diagnostic test”, what do you mean? Diagnosis of what?

Line 60: I’m not following you here. What “specific motor performance”? Please, elaborate further.

Lines 69–74: What are the measuring units of the burpee? Time to complete the burpee? Please explain to readers (and review board).

Line 77: I suggest removing “The authors state that…” and begin the sentence with “The HIIT…”

Line 92: This rationale fits better in the Methods section.

Lines 96–98: I would leave this to the Methods section as well.

Line 100: For simplicity purposes, clearly state the study objective(s).

Methods

Line 114 and forth: Participants is a more ethical name than test subjects.

Line 115: Report your sample size calculation.

Line 116: Isn’t the Ruffier test an exercise test? Me and other potential readers will not understand why you are then referring to resting HR. Describe succinctly the test to readers, when it was measured and how was HR measured. Also, you don’t refer readers to Table 1 at any time.

Line 117: I would add this “…to obtain the degree in Physical Education” after “As part of…”, i.e., “As part of the educational process to obtain the degree in Physical Education, the burpee exercise…”

Lines 119–127: The inclusion and exclusion criteria are not clear. Please report all of them.

Table 1: Define the abbreviations at the end of table. Also, inform how height and weight were measured, including instruments, and how BMI was calculated.

Line 130: remove “In terms of time,…” and begin the sentence with “The test is identical…”

Lines 131-142: I’m confused. Were participants instructed to perform burpees as fast as they can on each testing period (3 minutes) or to follow the pace of 7 seconds per burpee? Please, explain. It is not clear. Also, presenting pictures or drawings of each phase of the burpee, including when the measurements are being performed in a participant would be of great value for readers.

Line 145: state the developer of the PHYPHOX mobile app

Line 152: state de developer of Excel.

Line 153: state the manufacturer of SUUNTO device.

Lines 161–162: “The diagnostic series started with a barbell weight of 20 kg, which was gradually increased by 5 kg”. That’s it? Until 25 kg?

Line 181: “all.” is not correctly spelled.

Line 184: State the manufacturer of the Tendo…

Line 187: state the software used and the developer

Line 190: define the abbreviations

Line 201: I suggest separating Data Analysis into two different sections – Data processing and Data (or Statistical) analysis.

Line 202: I suggest using another abbreviation as “m” is the SI unit of length and distance.

Line 214: (Šiška, 2024) is not the citation format you have used previously.

Line 229: Explain why you use two different decision threshold values to reject H0.

Line 230: State the manufacturer/developer and define the acronyms not previously presented.

Results (also see my general comments)

Figure 2. Can you put the graphs all with the same longitudinal scale? It gives the impression that round III has more repetitions when it doesn’t. In the caption, tell readers what the dotted and solid lines and bars represent.

Line 250: It appears that you have performed one-way ANOVA but it was not expected in your statistical analysis section.

Line 251: Why performing post hoc tests if ANOVA had a p>.05?

Lines 254: it shouldn’t show. Most likely you have violations of the assumptions for performing ANOVA. Report the 95% confidence intervals for the mean differences.  Also, what post hoc test have you used? You are presenting F statistics for post hoc comparisons…

Line 262: Why using F statistics to compare 2 means?

Line 264: replace “is” by “was”.

Table 2 and 3: All abbreviations need to be defined at the end of the tables.

Figure 3: All abbreviations need to be defined.

Discussion (also see my general comments)

Line 287: just state burpees and remind readers on which indicators you are referring to.

Line 303: Reliability is a measurement property that you have not assessed in this study. I suggest replacing it by another term.

Line 315: You have not used reliability statistics. Please replace by another term.

Author Response

Response to Reviewer 1 Comments

1. Summary

2. Questions for General Evaluation

Reviewer’s Evaluation

Response and Revisions

Does the introduction provide sufficient background and include all relevant references?

Must be improved

The introduction was supplemented and expanded with other relevant information

Are all the cited references relevant to the research?

Yes/Can be improved/Must be improved/Not applicable

We checked the relevance of all cited references

Is the research design appropriate?

Yes

O.K.

Are the methods adequately described?

Must be improved

The methodology was expanded and described in more detail

Are the results clearly presented?

Must be improved

The results were appropriately processed and modified

Are the conclusions supported by the results?

Can be improved

We reconciled better conclusions and results

3. Point-by-point response to Comments and Suggestions for Authors

This is the reporting of a research studying the association between performance metrics of the Burpee Movement Program (BMP) and performance metrics of popular field strength/power and cardiorespiratory endurance measures. I find the article a little hard to read, either because authors are not clear when presenting their rationale, or because there are a lot of abbreviations that are not defined (e,g, GPS, HIIT), particularly in figures and tables in the Results section, hindering readability, or statistical analysis are not clear, e.g., results of ANOVAs are shown in Results but which models have been used, including which post hoc tests, have not been reported in Data Analysis section. Considering the study objectives, in addition, authors don’t clearly discuss the best correlations between BPM metrics and the other tests performance and why they best associate. They discuss a lot of correlations between other performance tests which is not their main study goal.

It was improved.

Title: Your title is a poorly informative. I suggest something like: “The Burpee Movement Program test and power and endurance performance measures: a cross-sectional correlation study in active young adults” or “The association between the Burpee Movement Program test and power and endurance performance measures in active young adults: a cross-sectional analysis”

The title was changed according to the recommendations.

Abstract: Clearly state your study objectives, i.e., refer to the burpee performance. Methods are incomplete. In results, if the Leger beep test is the 20m shuttle run test, just say is the 20m shuttle run test, otherwise you may confound several readers. Don’t use alternative names unless you clearly inform the reader on that (even so, avoid it for simplicity purposes).

The abstract was changed according to the comments and the leger beep test was replaced by the 20m shuttle run test (20mSR).

Introduction

Line 46: Cite your studies accordingly in the following sentences. Otherwise, it appears that you are only self-citing your studies.

It was improved.

Line 59: When you refer “diagnostic test”, what do you mean? Diagnosis of what?

The information was added.

Line 60: I’m not following you here. What “specific motor performance”? Please, elaborate further.

We decided to remove that paragraph, as it is not essential information in the context of the article.

Lines 69–74: What are the measuring units of the burpee? Time to complete the burpee? Please explain to readers (and review board).

Explanatory terms were added to the text.

Line 77: I suggest removing “The authors state that…” and begin the sentence with “The HIIT…”

It was removed

Line 92: This rationale fits better in the Methods section.

It was deleted. 

Lines 96–98: I would leave this to the Methods section as well.

It was removed and explained in method section. 

Line 100: For simplicity purposes, clearly state the study objective(s).

Objectives were added.

Methods

Line 114 and forth: Participants is a more ethical name than test subjects.

It was changed

Line 115: Report your sample size calculation.

This information was added to the participant section.

Line 116: Isn’t the Ruffier test an exercise test? Me and other potential readers will not understand why you are then referring to resting HR. Describe succinctly the test to readers, when it was measured and how was HR measured. Also, you don’t refer readers to Table 1 at any time.

We decided to remove the section on the Ruffier test, as it is not essential information in the context of the article.

Line 117: I would add this “…to obtain the degree in Physical Education” after “As part of…”, i.e., “As part of the educational process to obtain the degree in Physical Education, the burpee exercise…”

It was added

Lines 119–127: The inclusion and exclusion criteria are not clear. Please report all of them.

This information was added to the participant section.

Table 1: Define the abbreviations at the end of table. Also, inform how height and weight were measured, including instruments, and how BMI was calculated.

This information was added to the participant section.

Line 130: remove “In terms of time,…” and begin the sentence with “The test is identical…”

It was removed

Lines 131-142: I’m confused. Were participants instructed to perform burpees as fast as they can on each testing period (3 minutes) or to follow the pace of 7 seconds per burpee? Please, explain. It is not clear. Also, presenting pictures or drawings of each phase of the burpee, including when the measurements are being performed in a participant would be of great value for readers.

We added an explanatory sentence to the text as well as a picture representing the individual phases of single burpee repetition.

Line 145: state the developer of the PHYPHOX mobile app

It was stated

Line 152: state de developer of Excel.

It was stated

Line 153: state the manufacturer of SUUNTO device.

It was stated

Lines 161–162: “The diagnostic series started with a barbell weight of 20 kg, which was gradually increased by 5 kg”. That’s it? Until 25 kg?

This paragraph was improved.

Line 181: “all.” is not correctly spelled.

It was changed

Line 184: State the manufacturer of the Tendo…

It was stated

Line 187: state the software used and the developer

It was stated

Line 190: define the abbreviations

It was defined

Line 201: I suggest separating Data Analysis into two different sections – Data processing and Data (or Statistical) analysis.

It was changed

Line 202: I suggest using another abbreviation as “m” is the SI unit of length and distance.

It was changed

Line 214: (Šiška, 2024) is not the citation format you have used previously.

It was changed

Line 229: Explain why you use two different decision threshold values to reject H0.

It was changed to significance level p < .05

Line 230: State the manufacturer/developer and define the acronyms not previously presented.

It was stated

Results (also see my general comments)

Figure 2. Can you put the graphs all with the same longitudinal scale? It gives the impression that round III has more repetitions when it doesn’t. In the caption, tell readers what the dotted and solid lines and bars represent.

The graph was redrawn with the same longitudinal scale in all three rounds and explanatory notes were included in the text.

Line 250: It appears that you have performed one-way ANOVA but it was not expected in your statistical analysis section.

It was added to data analysis part. 

Line 251: Why performing post hoc tests if ANOVA had a p>.05?

Information about post hoc test was deleted.

Lines 254: it shouldn’t show. Most likely you have violations of the assumptions for performing ANOVA. Report the 95% confidence intervals for the mean differences.  Also, what post hoc test have you used? You are presenting F statistics for post hoc comparisons…

Information about post hoc test was deleted.

Line 262: Why using F statistics to compare 2 means?

It was one-way ANOVA notation according to apa style and the information was added to the data analysis section.

Line 264: replace “is” by “was”.

It was replaced

Table 2 and 3: All abbreviations need to be defined at the end of the tables.

Abbreviations were defined.

Figure 3: All abbreviations need to be defined.

Abbreviations were defined.

Discussion (also see my general comments)

Line 287: just state burpees and remind readers on which indicators you are referring to.

It was improved

Line 303: Reliability is a measurement property that you have not assessed in this study. I suggest replacing it by another term.

It was replaced by the term consistency

Line 315: You have not used reliability statistics. Please replace by another term.

It was replaced by the term consistency

Reviewer 2 Report

Comments and Suggestions for Authors

Thank you for inviting me to review this manuscript for your journal. The article by Šiska et al. is related to the use of accessible technology (mobile phone) and the validation of new tests for fitness evaluation. Despite the interesting nature of the research topic and its potential practical applications, in the opinion of this reviewer, there are several aspects that need to be better clarified:

1. Although each author’s narrative style is unique, the introductory and discussion chapters are filled with references to previous work by the same research group (which have little to do with the current study and are more of a historical review of their research) and references to variables not included in the present study. I encourage the authors to stick to their research question and focus on the associations between BMP and aerobic and strength performance, eliminating all narrative not related to the variables of their study, for the sake of clarity.

2. Instead of describing an “English squat,” I suggest the authors include a figure with the key positions of the mentioned burpee technique.

3. Fatigue index may not be calculated as the performance drop difference of the maximum and minimum values but from the difference between the first (or the maximun) and the last repetitions.

4. In this sense, only the last 4 of the 78 repetitions appear to have been under fatigue conditions, which could indicate a flaw in the selected protocol (not sufficiently demanding for the selected population) or a poor sensitivity of the monitoring device.

5. A calculation of the sample size or the power of the study is lacking.

5. The first paragraph of the discussion can briefly recall the objective of the study and then highlight the main findings of the study, not become a niche in which to cite previous related work of the authors.

6. Plase remove all references to sprint performance, Cooper test, and in general, anything not directly related with your hypothesis in the core of the discussion section, for the shake of clarity.

7. Please avoid using acronyms in the abstract, especially if they have not been previously defined (e.g., BMP).

8. Line 66, 80, 91,...: Plese remove the citations from the text

9. Line 105: Please be consistent in the use (or not) of the acronym BMP. 

10. Line 335: You can not state your hypothesis at this point. It should be clarify at the end of the introduction.

Comments on the Quality of English Language

Needs a review

Author Response

Response to Reviewer 2 Comments

1. Summary

2. Questions for General Evaluation

Reviewer’s Evaluation

Response and Revisions

Does the introduction provide sufficient background and include all relevant references?

Must be improved

The introduction was widen and fulfilled with some more information

Are all the cited references relevant to the research?

Yes/Can be improved/Must be improved/Not applicable

In the article all cited references are relevant to the research

Is the research design appropriate?

Must be improved

We supplemented and revised the research design

Are the methods adequately described?

Can be improved

We expanded and supplemented research methods

Are the results clearly presented?

Must be improved

The presentation of the results was revised in an effort to clarify them

Are the conclusions supported by the results?

Must be improved

The conclusions have been modified to reflect the results more

3. Point-by-point response to Comments and Suggestions for Authors

Thank you for inviting me to review this manuscript for your journal. The article by Šiska et al. is related to the use of accessible technology (mobile phone) and the validation of new tests for fitness evaluation. Despite the interesting nature of the research topic and its potential practical applications, in the opinion of this reviewer, there are several aspects that need to be better clarified:

1. Although each author’s narrative style is unique, the introductory and discussion chapters are filled with references to previous work by the same research group (which have little to do with the current study and are more of a historical review of their research) and references to variables not included in the present study. I encourage the authors to stick to their research question and focus on the associations between BMP and aerobic and strength performance, eliminating all narrative not related to the variables of their study, for the sake of clarity.

It was improved.

2. Instead of describing an “English squat,” I suggest the authors include a figure with the key positions of the mentioned burpee technique.

We added a picture representing the individual phases of one Burpee repetition to the manuscript.

3. Fatigue index may not be calculated as the performance drop difference of the maximum and minimum values but from the difference between the first (or the maximun) and the last repetitions.

It was explained in the discussion.

4. In this sense, only the last 4 of the 78 repetitions appear to have been under fatigue conditions, which could indicate a flaw in the selected protocol (not sufficiently demanding for the selected population) or a poor sensitivity of the monitoring device.

That was a rare case, but it is true that for some participants it was not sufficiently demanding.

5. A calculation of the sample size or the power of the study is lacking.

This information was added to the participant section

5. The first paragraph of the discussion can briefly recall the objective of the study and then highlight the main findings of the study, not become a niche in which to cite previous related work of the authors.

It was improved.

6. Plase remove all references to sprint performance, Cooper test, and in general, anything not directly related with your hypothesis in the core of the discussion section, for the shake of clarity.

The references and text were removed

7. Please avoid using acronyms in the abstract, especially if they have not been previously defined (e.g., BMP).

Acronym BMP was deleted and in the end of abstract was replaced with full title.

  1. Line 66, 80, 91,...: Plese remove the citations from the text

Citations have been deleted.

  1. Line 105: Please be consistent in the use (or not) of the acronym BMP. 

It was improved throughout the manuscript

  1. Line 335: You can not state your hypothesis at this point. It should be clarify at the end of the introduction.

It was deleted.

Comments on the Quality of English Language

Needs a review

English was reviewed.

Round 2

Reviewer 1 Report

Comments and Suggestions for Authors

After reading responses and amendments performed by the authors, readability has significantly been improved in this version of the manuscript, particularly, in the Discussion section. However, I find that Discussion section is not following, at least partially, the rationale of the Introduction or Abstract. After reading again this whole article, it seems that authors were searching more of what type of workout regimen could be the BMP categorized into and poorly on the diagnosis of physical abilities of a person. My first impression was that they were studying the BPM as a diagnostic tool for physical capacities, and they were testing validity, as a measurement property. If they were, in fact, looking for what they conclude in lines 358 to 365, Introduction needs to be rewritten to more clearly guide readers. Remove all diagnoses wording, including accuracy or test or tool (check the title again; abstract is also misleading readers) and simplify the Introduction. There are a lot of description of other studies, including theirs, but the rationale (with point and counter-point perspectives), study problem and hypothesis are not clear. They should use the findings of those studies to support their rationale and hypothesis.

Although the reporting has been significantly improved, there are important issues that need to be solved. One-way ANOVA treats groups has independent from each other which is not the case here. They have only one group, hence the (less) variability of variance is not accounted for when comparing different time points (BPM performance) or different tests. A within-subject/repeated measures design corrects the variance that I report here. They also report they will present r square statistics, but r square statistics are never reported in the Results section. 

Title: remove “test”

Abstract:

Lines 27–28: You have not used predictive statistical analysis, such as, linear regression analysis. 

Line 59: Improve this sentence.

Line 62: provide examples of such “certain number of values”

Lines 83: Is fatigue index this calculated from the average of all burpees intensities? You are making the reader go to all your previous studies to understand your descriptions and reasonings. You must provide sufficient information so that they don’t have to read them all. As it is, certainly, many readers will abandon further reading this manuscript.

Lines 85–100: Why is HIIT refer here again? You believe that, conceptually, the BMP could be included as a HIIT workout? (line 55) Are you trying to figure out if it is? Or it may relate to other physical characteristics or abilities? Do you see the potential of this way of assessment physical capacity a new tool or replace others? You are describing things, results of yours and other studies, but you are not connecting the dots to readers.

Lines 101_102: What do you mean by accuracy? You have not performed diagnostic accuracy analysis. I suggest replacing by another term.

Line 108: You have not explained to readers clearly what the fatigue index is and how it is measured. Lines 79 to 84 do not provide sufficient information on that.

Line 244: 1,2,3??

Line 248: What about homogeneity of variance?

Line 248_250: This is not correct. You have only one population. One-way treats the variance as if the groups are from different populations. 

Line 372: You have not used predictive statistical analysis to conclude this. Even if ANOVA can be used as such a tool you have not selected the model correctly. R square can give a sign of prediction but you don’t present it.

Author Response

Comments and Suggestions for Authors

After reading responses and amendments performed by the authors, readability has significantly been improved in this version of the manuscript, particularly, in the Discussion section. However, I find that Discussion section is not following, at least partially, the rationale of the Introduction or Abstract. After reading again this whole article, it seems that authors were searching more of what type of workout regimen could be the BMP categorized into and poorly on the diagnosis of physical abilities of a person. My first impression was that they were studying the BPM as a diagnostic tool for physical capacities, and they were testing validity, as a measurement property. If they were, in fact, looking for what they conclude in lines 358 to 365, Introduction needs to be rewritten to more clearly guide readers. Remove all diagnoses wording, including accuracy or test or tool (check the title again; abstract is also misleading readers) and simplify the Introduction. There are a lot of description of other studies, including theirs, but the rationale (with point and counter-point perspectives), study problem and hypothesis are not clear. They should use the findings of those studies to support their rationale and hypothesis.

Although the reporting has been significantly improved, there are important issues that need to be solved. One-way ANOVA treats groups has independent from each other which is not the case here. They have only one group, hence the (less) variability of variance is not accounted for when comparing different time points (BPM performance) or different tests. A within-subject/repeated measures design corrects the variance that I report here. They also report they will present r square statistics, but r square statistics are never reported in the Results section. 

We tried to improve the text according to the recommendations. We have simplified the introduction, improved the methodological part and the discussion.

Title: remove “test”

It was removed

Abstract:

Lines 27–28: You have not used predictive statistical analysis, such as, linear regression analysis. 

It was replaced by the word potential 

Line 59: Improve this sentence.

The sentence was improved

Line 62: provide examples of such “certain number of values”

It was improved

Lines 83: Is fatigue index this calculated from the average of all burpees intensities? You are making the reader go to all your previous studies to understand your descriptions and reasonings. You must provide sufficient information so that they don’t have to read them all. As it is, certainly, many readers will abandon further reading this manuscript.

It was explained

Lines 85–100: Why is HIIT refer here again? You believe that, conceptually, the BMP could be included as a HIIT workout? (line 55) Are you trying to figure out if it is? Or it may relate to other physical characteristics or abilities? Do you see the potential of this way of assessment physical capacity a new tool or replace others? You are describing things, results of yours and other studies, but you are not connecting the dots to readers.

It was rewritten

Lines 101_102: What do you mean by accuracy? You have not performed diagnostic accuracy analysis. I suggest replacing by another term.

Word accuracy was deleted

Line 108: You have not explained to readers clearly what the fatigue index is and how it is measured. Lines 79 to 84 do not provide sufficient information on that.

It was explained

Line 244: 1,2,3??

It was explained

Line 248: What about homogeneity of variance?

It was replaced by a paired samples t test

Line 248_250: This is not correct. You have only one population. One-way treats the variance as if the groups are from different populations. 

It was replaced by a paired samples t test

Line 372: You have not used predictive statistical analysis to conclude this. Even if ANOVA can be used as such a tool you have not selected the model correctly. R square can give a sign of prediction but you don’t present it.

It was improved

Reviewer 2 Report

Comments and Suggestions for Authors

I would like to thank the authors for the revisions made to your manuscript. It is clear that several of the suggested changes have been addressed effectively, enhancing the overall quality of the work.

However, some issues persist, particularly the references to previous work by the same research group, which somewhat cloud the scientific rigor of the study. Although the effects of BMP on fatigue may be minimal and the overall scientific impact of the publication is affected by this, I leave the final decision in the hands of the editors.

Author Response

Comments and Suggestions for Authors

I would like to thank the authors for the revisions made to your manuscript. It is clear that several of the suggested changes have been addressed effectively, enhancing the overall quality of the work.

However, some issues persist, particularly the references to previous work by the same research group, which somewhat cloud the scientific rigor of the study. Although the effects of BMP on fatigue may be minimal and the overall scientific impact of the publication is affected by this, I leave the final decision in the hands of the editors.

We tried to improve the text according to the recommendations. In the discussion, we tried to clarify the contribution of the fatigue index.